# A new adaptive MPPT technique using an improved INC algorithm supported by fuzzy self-tuning controller for a grid-linked photovoltaic system

**Nagwa F. Ibrahim**[1], **Mohamed Metwally Mahmoud**[2], **Hashim Alnami**[3], **Daniel Eutyche Mbadjoun Wapet**[4]*, **Sid Ahmed El Mehdi Ardjoun**[5], **Mohamed I. Mosaad**[6,7], **Ammar M. Hassan**[8], **H. Abdelfattah**[1]

**1** Electrical Department, Faculty of Technology and Education, Suez University, Suez, Egypt, **2** Department of Electrical Engineering, Faculty of Energy Engineering, Aswan University, Aswan, Egypt, **3** Electrical Engineering Department, Jazan University, Jazan, KSA, **4** National Advanced School of Engineering, Université de Yaoundé I, Yaoundé, Cameroon, **5** IRECOM Laboratory, Faculty of Electrical Engineering, Djillali Liabes University, Sidi Bel-Abbes, Algeria, **6** Electrical & Electronics Engineering Technology Department, Yanbu Industrial College (YIC), Royal Commission Yanbu Colleges & Institutes, Yanbu, Saudi Arabia, **7** Electrical Engineering Department, Faculty of Engineering, Damietta University, Damietta, Egypt, **8** Arab Academy for Science, Technology and Maritime Transport, South Valley Branch, Aswan, Egypt

* eutychedan@gmail.com

**Data Availability Statement:** All relevant data are within the paper.

## Abstract

Solar energy, a prominent renewable resource, relies on photovoltaic systems (PVS) to capture energy efficiently. The challenge lies in maximizing power generation, which fluctuates due to changing environmental conditions like irradiance and temperature. Maximum Power Point Tracking (MPPT) techniques have been developed to optimize PVS output. Among these, the incremental conductance (INC) method is widely recognized. However, adapting INC to varying environmental conditions remains a challenge. This study introduces an innovative approach to adaptive MPPT for grid-connected PVS, enhancing classical INC by integrating a PID controller updated through a fuzzy self-tuning controller (INC-FST). INC-FST dynamically regulates the boost converter signal, connecting the PVS's DC output to the grid-connected inverter. A comprehensive evaluation, comparing the proposed adaptive MPPT technique (INC-FST) with conventional MPPT methods such as INC, Perturb & Observe (P&O), and INC Fuzzy Logic (INC-FL), was conducted. Metrics assessed include current, voltage, efficiency, power, and DC bus voltage under different climate scenarios. The proposed MPPT-INC-FST algorithm demonstrated superior efficiency, achieving 99.80%, 99.76%, and 99.73% for three distinct climate scenarios. Furthermore, the comparative analysis highlighted its precision in terms of control indices, minimizing overshoot, reducing rise time, and maximizing PVS power output.

**Funding:** The author(s) received no specific funding for this work.

**Competing interests:** The authors have declared that no competing interests exist.

**Abbreviations:** D, Duty cycle; DN, Duty cycle negative value; DNV, Duty cycle very negative value; DP, Duty cycle positive value; DPV, Duty cycle very positive value; DZ, Duty cycle zero value; FL, Fuzzy logic; FST, Fuzzy self-tuning; INC-ST, Incremental conductance-Fuzzy self-tuning; $I_{ph}$, photo-generated current; K, Boltzmann gas constant; $K_D$, Derivative gain; $K_I$, Integral gain; $K_P$, Proportional gain; m, Diode quality factor; N, Negative value; NV, Very negative value; P, Positive value; PV, Photovoltaic; PV, Very positive value; $R_s$, series resistor; $R_{sh}$, shunt resistor; Tc, Cell absolute temperature; Z, Zero value; $\Delta$I, Change in current; $\Delta$P, Change in power; $\Delta$V, Change in voltage.

# 1. Introduction

## a) Motivation

Given the current economic crises, which have increased in severity with Covid 19, it has become imperative to integrate renewable energy sources (RESs) into the power grid and maximize the utilization of the available RESs to alleviate these crises [1, 2]. A future for green power is being offered by the rising consumption of RESs, which is bringing down costs. RESs such as wind and PV systems (PVSs) are the driving forces behind this sustainable energy transition. It is acknowledged that the power produced by the PVSs is secure and readily accessible on Earth every day. 1.7% of the globe's electricity is currently supplied by PVSs, and by 2025, the output power should approach 1 TW [3]. The extensive integration of PVSs is due to several advantages of PVSs, including maintenance facilities and being environmentally friendly [4]. In addition, PVSs are characterized by environmental credentials, free energy sources, high efficiency, low cost, electricity generation without moving parts, and longevity compared to other RESs. PVSs may be categorized as autonomous and connected to electrical energy systems. In autonomous PVSs, a battery bank is required to store the energy in periods of unavailability of power from the PVSs. This application is applicable for low-power applications. In contrast, grid-connected PVSs do not require any storage element and they are suitable for high-power applications, and on-grid systems [5, 6].

The main elements that limit the quantity of electrical energy that could be captured include atmospheric parameters, dirt, temperature (T), rainfall, cloud cover, and geography. Because of the radiation's varying angle of azimuth per hour, the level of irradiance (I) usually varies during the daytime. Any PVS produces different amounts of electricity based on the load of the panel, which reduces the quantity of electricity generated even at the same I and T. The maximum power point (MPP) at a particular load occurs and changes throughout the course of a day with Ts and I variations, rendering it difficult to locate and compute the MPP [7]. The primary emphasis of the work is MPPT, not other PVS difficulties. The output voltage of the PV system is DC which is connected to the grid through a DC-DC power converter (PC) and DC-AC PC. Buck-boost, boost, and buck power converters (PCs) are three of the main MPPT architectures PVS uses to track the MPP. This PC is used to adjust the DC's PVS voltage by regulating the duty cycle (D) to a level where the MPP is achieved by different control algorithms. The boost PC is perfect for PV uses due to its tiny switching losses as well as low inductivity, which reduces current ripple. Additionally, this PC operates with a stable current and less current stress than previous architectures. On the contrary, the DC-AC PC is utilized to link the output of the DC-DC PC to the grid [8–10].

## b) Background

MPPT systems require a method of control to be able to boost their efficacy. Because of its ease of use and versatility in execution, the PID controller is frequently used in MPPT devices. With its straightforward framework, PID, nevertheless, exhibits a low efficiency for MPPT uses [11]. The most-viewed techniques are the conventional techniques (CTs), like INC. CTs could only monitor the MPP when the weather was steady, notwithstanding their straightforward design and use. MPPT CTs are also inefficient for large-scale PVSs and show instabilities close to the MPP. Owing to the abovementioned limitations, scientists and engineers from all over the world are creating new strategies for managing MPPT [12, 13].

A number of the latest and most significant modern MPPT methods that potentially address some of the problems brought on by conventional MPPT controllers include soft calculation (SC), artificial intelligence (AI), and bio-inspired technology (BT) [14, 15]. Sophisticated

methods for MPPT possess an outstanding ability for monitoring the MPP despite their high level of complexity. Among the most popular advanced MPPT algorithms include heuristic techniques, including particle swarm optimization (PSO), fuzzy logic (FL), and neural networks (NN). One of the most effective methods for tackling nonlinear issues is the MPPT procedure, which relies on SC [16]. Unluckily, compared to CTs, these methods of MPPT are costly to implement, require an accurate learning information set, and are more complicated.

In these studies, various advanced algorithms and techniques are presented to optimize power generation in PVSs. They include the use of FL based on particle swarm optimization in [17], adaptive neuro-FL inference systems shown in [18], and Lyapunov controllers in [19], each tailored to achieve efficient MPPT while enhancing power quality and grid integration as shown in [20, 21]. These approaches minimize oscillations and reduce harmonic distortions, ultimately improving performance under varying environmental conditions and demonstrating innovative solutions for the renewable energy field.

## c) Literature review

There is a wealth of research available in the literature aimed at addressing the drawbacks of existing techniques and trying to enhance them. PVSs are suggested for a unique FL-based mix MPPT strategy. To calculate the MPP, an offline current from a short circuit is employed, and the FL is then applied to get the precise magnitude of the highest possible power. The findings show that, in a range of climatic circumstances, the suggested method surpassed the combination technique (open-circuit voltage and P&O methodology) [22]. A revolutionary P&O strategy is put into practice and optimized for MPPT using NN software. To confirm the system's operation under various levels of I, studies were carried out. The suggested study shows that the NN-optimized P&O strategy surpasses the conventional INC techniques under different levels of I and Ts. It has been demonstrated that this device can generate roughly 99% of the actual maximal power. The NN technique only needs about 0.025 s to attain the target value, with minimal overshoot, as opposed to INC, which needs roughly 0.3 s [23]. FL was used to quickly and effectively design the membership features for MPPT with no the aid of a certified specialist. The FL was significantly superior to the NN-PSO, NN-GA, and NN-imperialist competitive method (ICM) in terms of rigidity, precision, rapidity, and simple deployment versus climatic perturbations [16]. A type 2 FL set and a super-twisting sliding mode controller (STSMC) (STSMC-T2FC) were developed to address the chattering problem. The STSMC-T2FC MPPT's efficacy is 99.59%, whereas STSMC's and SMC's are 99.33% and 99.20%, respectively. The efficiency performances were close, but STSMC-T2FC won out [24]. A boost PC-based reliable direct adaptive controller was created for MPPT. MATLAB/Simulink is used to verify the controller's dependability under various operating situations after an analytical model was created and an appropriate technique was created for the MPPT [25]. To make certain that all of the PV's output is transferred to the load, a two-step globe MPPT control technique was proposed. To find the universal MPP, the first step uses global perturbation-based extremum-seeking control. MRAC, which is utilized to control the DC-DC PC dynamics, is the second step. The simulation evaluates the performance of the suggested controller in terms of tracking speed, efficiency, and accuracy under various radiation situations [26]. A method was developed employing the adapted MRAC combined with the Lyapunov and INC technique to rapidly obtain MPP in the face of variations in I and T. Tests showed that the suggested technique tracks the MPP in comparison to the INC-PI controller [27]. In different circumstances, an MPPT algorithm for the PVS was offered that makes use of an ANFIS. P&O and FL cannot follow the MPP as quickly as the suggested technique can in erratic climates [28].

The typical MPPT techniques, fractional open-circuit voltage (FOCV), and fractional short-circuit current (FSCC) were proposed in many MPPT applications, however, they did not demonstrate high efficiency and accuracy in reaching the MPP [29]. Ref. [30] used a P&O algorithm-based MPPT charge controller for a stand-alone 200 W PV system. Despite the P&O algorithm being a straightforward strategy and simple to implement, the transient and steady-state operations are highly affected by the perturb step [31]. The key solution to this perturbs step was handled by applying adaptive regulation of the step, which is a high computational burden. The INC method was presented to overcome the issues that arise while employing the P&O MPPT technique. In transient periods with rapid changes in the environment, INC beat P&O. The difficulty in adapting the INC technique is updating the controller parameters in response to changes in the environment [32].

The PID controller, which is widely used in numerous MPPT approaches as well as other RES applications, is the most simple [33]. Certain advancements in the INC-MPPT technique were made by adaptively modifying the PID controller parameters, which in turn affected the D of the DC-DC converters. Finding the PI controller parameters and quickly adjusting them to follow environmental changes was one of the challenges in these advancements [34]. Due to the nonlinearity of the grid-connected PV systems and the uncertainty of the PV-generated power due to climatic fluctuations, setting the PID controller parameters for MPPT is a challenging issue. This challenge necessitated some adjustments and adaptations to the PID controller-based MPPT approaches [35, 36]. Some other modifications were presented to improve the INC technique. Ref. [35] presents coordinated MPPT and voltage regulation control using a one-step large gain DC-DC PC in a grid-linked PVS. MPPT fuzzy controller for PVSs using an FPGA circuit was investigated in [37]. Implementation of a reworked plan INC-MPPT algorithm with direct control based on a fuzzy D change estimator using dSPACE was introduced in [38]. ANN was utilized to update the D control signal in order to accomplish MPPT in a PV system. Despite the advantages of adopting ANN over static PID controllers, they were more sophisticated [39].

According to a cursory review of the scientific literature, CTs are simple to set up and yield beneficial outcomes for MPPT, but they have the problem of exhibiting unfavorable fluctuations close to MPP. However, they are difficult, costly, and time-consuming to use, SC techniques are the most effective at monitoring MPPT. Consequently, this inspired the author to create an adaptive regulator that may mitigate the shortcomings of CT and SC methods. It is therefore extremely difficult to significantly enhance the MPPT technique's capabilities in terms of level of complexity, monitoring rapidity, precision, oscillations around MPP, and monitoring effectiveness in changing circumstances.

## d) Contributions

The paper introduces an innovative INC-FST to improve MPPT efficiency, which will reduce system management complexities and effectively handle instabilities and disturbances in the investigated PVS. This work improves the INC-MPPT algorithm with a fuzzy self-tuning-based PID controller (INC-FST) for regulating the MPPT control of the studied system. This self-tuning algorithm will drive the DC-DC PC D to be updated to track the MPP with changes in I and T. The suggested method is simple, has higher dynamic responsiveness, barely oscillates near MPP, tracks quickly, and performs better in variable weather circumstances. The MATLAB/Simulink is used to investigate the proposed technique. Three tests are applied (variable I with constant T (C1), variable T with constant I (C2), and variable I with variable T (C3)) to assess the proposed effectiveness. The suggested technique efficacy under C1, C2, and C3 are 99.80%, 99.76%, and 99.73%, respectively. The PVS output, including power, voltage,

current, and DC bus voltage, shows that the proposed technique performs admirably under a variety of conditions, including I and T fluctuations. The key finding of the suggested work is summarised as follows:

- For PVSs to enable effective MPPT, an innovative INC-FST is suggested.

- The suggested controller has low instabilities around MPP, excellent efficacy, faster rapid responsiveness, and quick converging time.

- The INC-FST is resistant to I and T change since it is adaptable.

- The enhanced algorithm is contrasted with three control approaches (P&O, INC, and INC-FL controller). The improved INC algorithm outperformed previous algorithms in some control indices, such as overshoot, rise time, and settling time, and demonstrated a quicker dynamic reaction for MPPT.

### e) Organization

The rest of this paper is organized as follows: Section 2 gives a synopsis of the system under study and the PVS modeling. Section 3 demonstrates P&O and INC MPPT algorithms. Section 4 explains a modified INC algorithm with an INC-FST controller. The simulation's work findings are reported in Section 5. Finally, conclusions and discussions are presented in section 6.

## 2. System understudy and PV modeling

The grid-connected PVS consists of PV solar cells, PCs, filters, and grid interfaces control system as shown in Fig 1a and 1b. The system data is given in Tables 1 and 2.

The PV cell is the most main component of a PVS. A PV module is a group of connected cells to form a PV panel; many panels make up a PV array. Power generated by the PV generator is affected by T and I. The practical electrical circuit of the solar cell and the inverted diode is depicted in Fig 2 [14, 40].

The output current model by the solar cell can be expressed in Eq (1) as [41]:

$$I = I_{ph} - I_D - I_{sh} \tag{1}$$

Calculations for the diode's altered current based on the Shockley diode equation can be expressed as [8]:

$$I_D = I_O \left( \exp\left[\frac{q(V + IR_s)}{mkT_c}\right] - 1 \right) \tag{2}$$

The PV cell current can be determined as:

$$I = I_{ph} - I_O \left( \exp\left[\frac{q(V + IR_s)}{mkT_c}\right] - 1 \right) - \frac{V + IR_s}{R_{sh}} \tag{3}$$

In which m is the diode quality factor, V is the PV cell output voltage, Tc is the cell's absolute temperature in Kelvin, Rs is the cell's series resistance in Ohms, and $R_{sh}$ is the cell's shunt resistance in Ohms, Io is the saturation current diode in ampere, q is the electron charge in coulomb and K is the Boltzmann gas constant. Both single-diode and double-diode types of PV modules are widely used today. Double-diode models can be used to correctly represent the solar panel. Because of its simplicity and accuracy, the single-diode model is utilized in this paper. The following equation can be used to express the current (Imp) at the point of

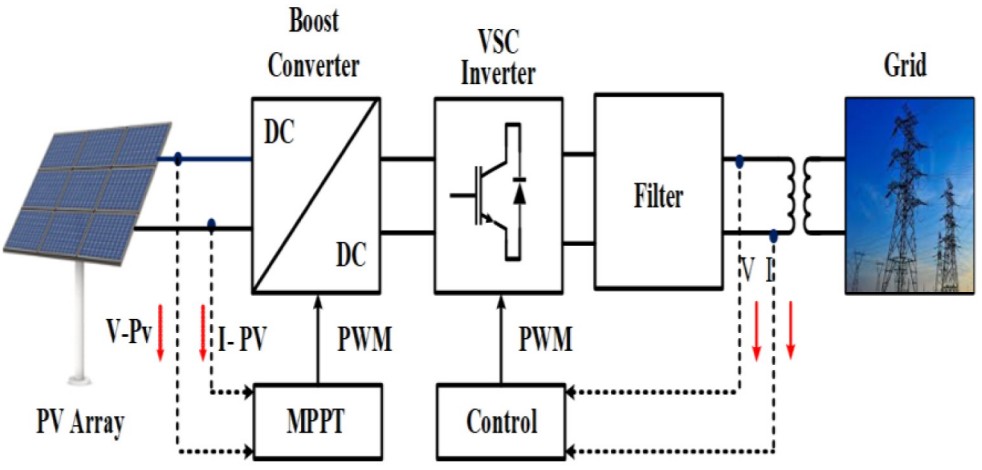

a-    Schematic diagram of the grid-tied PVS.

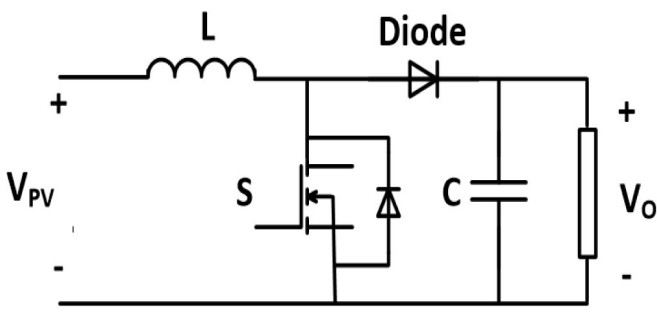

b-    DC-DC boost PC.

**Fig 1. System understudy.** (a) Schematic diagram of the grid-tied PVS. (b) DC-DC boost PC.

Table 1. Specifications of Solarex MSX60 60 W PV panel.

| Parameter | Value |
|---|---|
| Open circuit voltage ($V_{OC}$) | 64.2 V |
| Short circuit current ($I_{sc}$) | 5.96 A |
| Voltage at max power ($V_{m}$) | 54.7 V |
| Current at max power ($I_{m}$) | 5.58 A |
| Maximum power ($P_{m}$) | 305.226 W |
| Temperature coefficient of open circuit voltage ($\beta$) | -(0.27269) mV/°C |
| Temperature coefficient of short circuit current ($\alpha$) | (0.061745±0.015) %/°C |
| Temperature coefficient of power ($\mu$) | -(0.512±0.05) %/°C |
| Light-generated current (IL) | 6.0092 |
| Shunt resistance (Rsh) | 269.5934 Ω |
| Series resistance (Rs) | 0.37152 Ω |
| Frequency (F) | 50 HZ |
| X/R Ratio | 7 |
| Grid Voltage (RMS) | 500 V |

**Table 2. Parameters of DC\DC-PC.**

| Parameter | Value |
|---|---|
| DC link capacitor ($C_{dc}$) | 12 mF |
| Inductor (L) | 5 mH |
| Capacitor (C) | 100 pF |
| Resistor (R) | 0.005 Ω |

maximum power.

$$I_{mp} = I_{ph} - I_O \left( \exp \left[ \frac{q(V + IR_s)}{mkT_c} \right] - 1 \right) - \frac{V_{mp} + I_{mp}R_s}{R_{sh}} \tag{4}$$

However, the maximum power s (Pmax) is given by:

$$P_{max} = V_{mp} \left\{ I_{ph} - I_O \left( \exp \left[ \frac{q\left(V_{mp} + I_{mp}R_s\right)}{mkT_c} \right] - 1 \right) - \frac{\left(V_{mp} + I_{mp}R_s\right)}{R_{sh}} \right\} \tag{5}$$

where $I_{mp}$ is the maximum panel current and $V_{mp}$ is the maximum panel voltage.

The output of the PVS is connected to the grid through a boost DC\DC-PC and DC\AC-PC as given in Fig 1b. The boost PC is controlled to track the MPP from the PVS. The output and the input voltage to the boost PC ($V_O$, $V_{PV}$), respectively, are related as:

$$\frac{V_O}{V_{PV}} = \frac{1}{1 - D} \tag{6}$$

By adjusting the D, the output voltage of the boost PC is regulated to track the voltage at which the MPP is achieved.

## 3. Studied MPPT algorithms

The primary role of MPPT is to maximize the amount of energy from the PVS by adjusting the system operating voltage at the most efficient value corresponding to the MPP at different Ts and Is [42]. As the I increase at a constant T, the voltage, and current increase and,

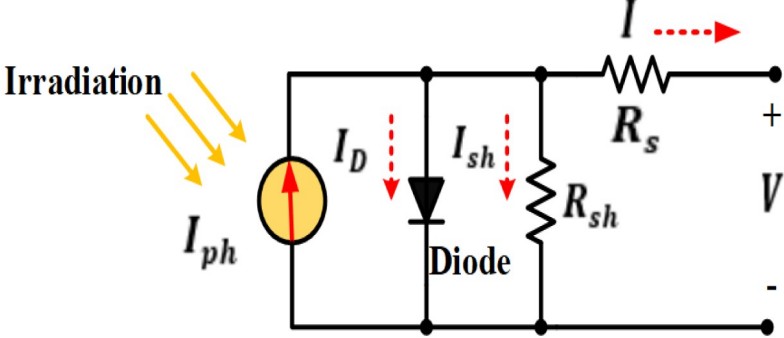

**Fig 2. Equivalent circuit of a PV cell.**

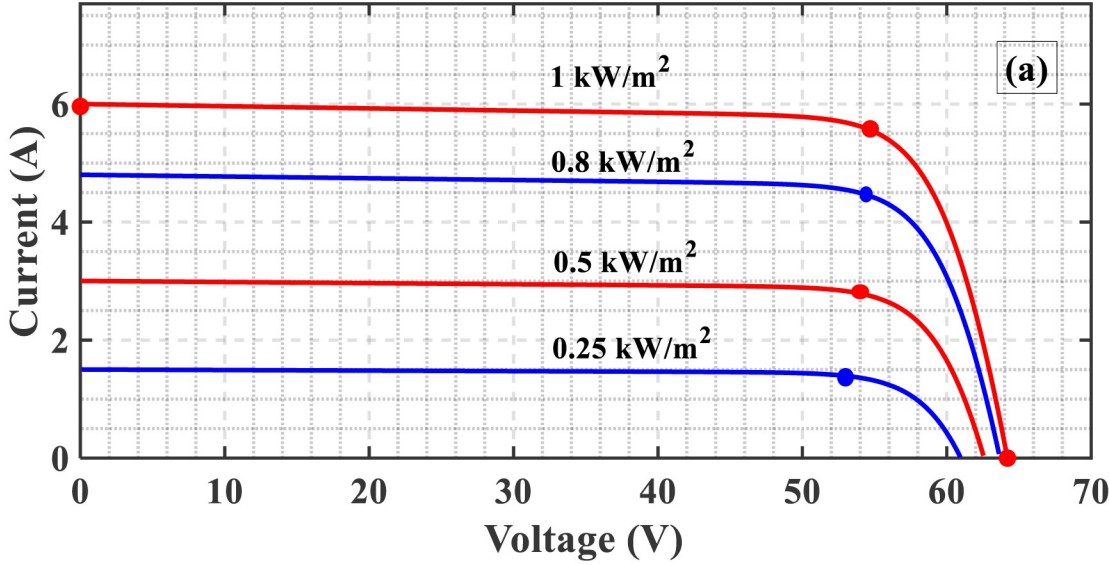

(a)  I-V characteristics.

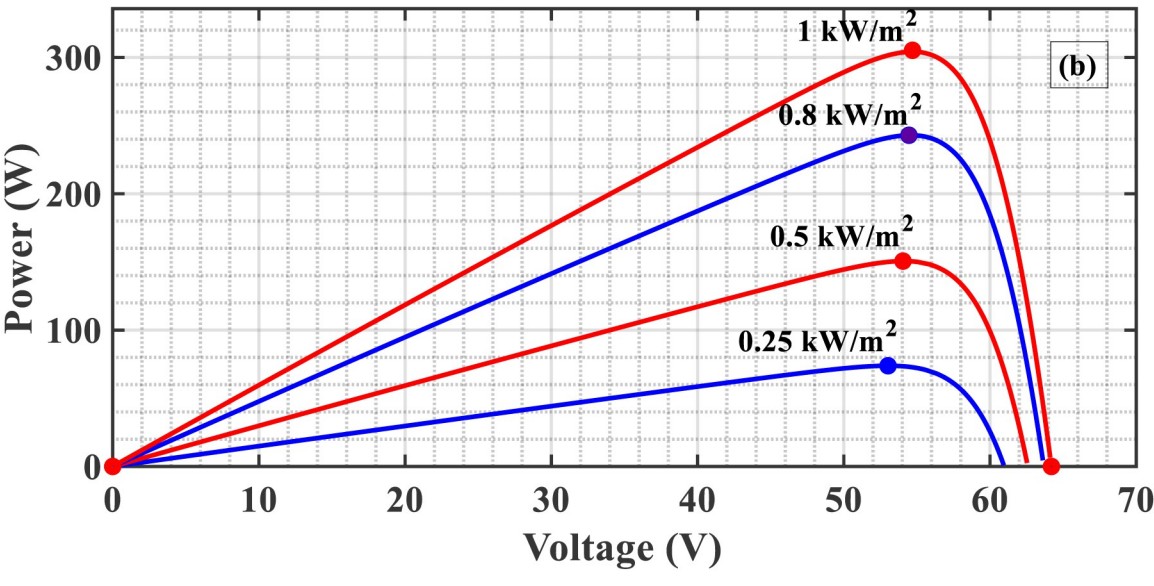

(b)  P-V characteristics.

**Fig 3. Features of the PVS as I change.** (a) I-V characteristics. (b) P-V characteristics.

consequently, the PV-generated power increases as shown in Fig 3a and 3b. If the T increases at constant I, the PV voltage is almost constant, and the current decreases. Consequently, the PV-generated power decreases, as depicted in Fig 4a and 4b. From these results, the PV-generated power is affected mainly by the values of the Is and Ts. In addition, the DC output voltage of the boost PC significantly affects the power extracted from the PVS hence controlling this DC voltage to the value at which MPP is used to achieve MPPT.

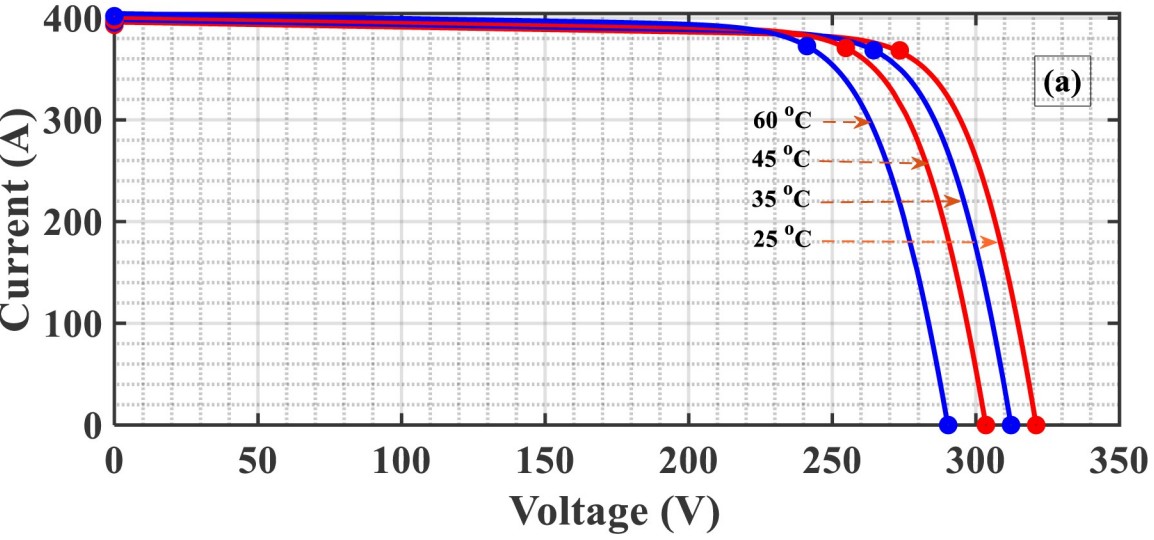

(a) I-V characteristics.

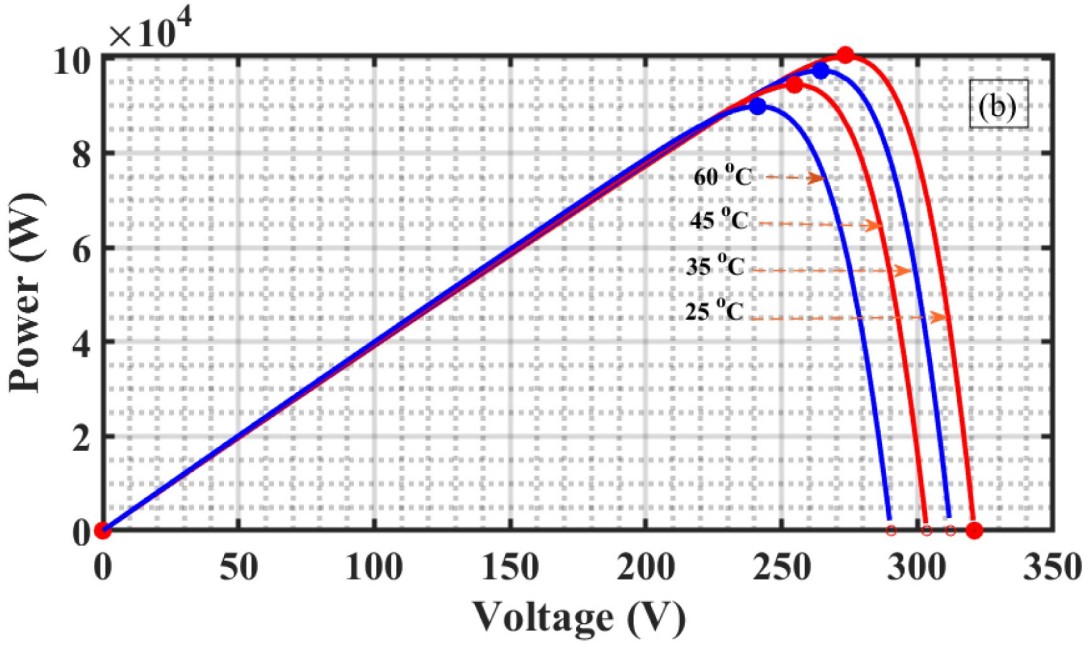

(b) P-V characteristics.

**Fig 4. Features of the PVS as T changes.** (a) I-V characteristics. (b) P-V characteristics.

The P&O algorithm is the most popular MPPT algorithm. This method has many benefits, including low cost, rapid deployment, fewer parameters, and the flexibility to make changes [31, 34]. The development of this method is based on research on the link between voltage and power production from solar modules [31, 34].

## 4. Improved INC algorithm with FL controller

### 4.1 INC algorithm

Incremental variations in PV array voltage and current are sensed by the controller in INC algorithms to estimate the impact of a voltage variation. To keep up with rapidly changing conditions, this technique requires more controller computing than the P&O algorithm [34, 37]. Like the P&O method, it can generate output power oscillations. The PV array's incremental conductance ($\Delta I/\Delta V$) is used to determine the direction of power change concerning voltage ($\Delta P/\Delta V$) in this technique. MPP is determined by comparing incremental conductance ($\Delta I/\Delta V$) to the array conductance (I/V) in the INC method. The output voltage is the MPP voltage if these two are equal ($\Delta I/\Delta V = I/V$). For as long as the I or T change occurs, the controller maintains this voltage. It is based on the fact that at maximum power, $\Delta P/\Delta V = 0$ and $P = VI$, hence the INC method is named after this fact [43]. The mathematical representation of the INC algorithm can be summarized thusly: The source's output power can be expressed as follows:

$$P = V*I \tag{7}$$

Normally, a source's output voltage is positive. Consequently, this algorithm's primary goal is to locate the voltage operating point where conductance equals incremental conductance. These ideas are expressed in Eq (8). The P–V curve slope is a key factor in determining the INC algorithm. The slope value at MPP is zero, grows (positive) on the left, and decreases (negative) on the right.

$$\begin{cases} \dfrac{\Delta P}{\Delta V} > 0, & \textit{Left side of the MPP} \\[2mm] \dfrac{\Delta P}{\Delta V} = 0, & \textit{at the MPP} \\[2mm] \dfrac{\Delta P}{\Delta V} < 0, & \textit{Right side of the MPP} \end{cases} \tag{8}$$

Some modifications in the INC have been presented to update the controller parameters in light of the changes in the environmental conditions to track the maximum power rapidly.

### 4.2 FL controller

FL control is a method for developing nonlinear controllers based on heuristic data derived from expert expertise, as illustrated in Fig 5, blue color. In this paper, a FL controller with two inputs and one output is created. The two input variables are error (E) and change (CE), which can be defined for sample times K:

$$E(K) = \frac{P(K) - P(K-1)}{V(K) - V(K-1)} = \frac{\Delta P}{\Delta V} \tag{9}$$

$$CE(K) = E(K) - E(K-1) = \Delta E \tag{10}$$

The slope of the P-V curve is the input $E(K)$, which determines where the MPP is in the PV module. The $CE(K)$ input determines whether or not the operating point is moving in the MPP direction. The increase in D is the output variable, which can take positive or negative values depending on where the operational point is located. To drive the load, this output is delivered to the DC-DC PC. An accumulator was used to calculate the D using the value of $D$

**Fig 5. Overall system with the INC-FL and INC-FST.**

provided by the controller.

$$D(K) = D(K-1) + \Delta D(K) \tag{11}$$

The main drawback of this FL controller is that the PID controller parameters are not updated with variations in the Ts and Is which will lead to a non-guarantee of extracting the maximum power from the PVS at these environmental variations.

The INC and INC-FL are not adaptive control methods, as inferred from the foregoing discussion. In other words, under specific environmental conditions, their controller parameters are adjusted. This signifies that the best operation is only provided under these conditions, and if they are changed, the grantee will not be able to reach the best operations. This calls for adaptive control techniques. These methods provide for optimal performance by allowing the

controller parameters and, in turn, the control signal, to be adjusted. These adaptive techniques surpluses the non-adaptive ones in many applications [34, 44]. Using INC-FST to update the control parameter and subsequently, the control signal, the adaptive control technique for MPPT improvement is provided in this study.

### 4.3 FL self-tuning

FL self-tuning algorithm for MPPT is developed to guarantee obtaining the greatest amount of possible power from the PVS by updating the controller parameters with any changes in the environmental conditions. The error and change of error are employed as inputs to the FL self-tuning while the PID controller gains ($k_{P1}$, $k_{I1}$, $k_{D1}$) are the outputs. The FL controller is added to the traditional PID controller to adjust the parameters ($k_P$, $k_I$, $k_D$) of the PID controller online according to the error along with the change in the error. As indicated in Fig 5, red color, the controller proposed that the operating ranges (discourse universe) be made more general to meet them.

The fixed gains of the PID controller are not updated with any changes in the T and I. Using a self-tuning FL controller to update the PID controller parameters according to the changes in Ts and Is to guarantee that the PVS is operating at MPP will be investigated.

Now the control action of the PID controller after self-tuning can be defined as:

$$u_{PID}(t) = k_{P2}\, e(t) + k_{I2} \int_0^t e(t)d(t) + k_{D2} \frac{de(t)}{dt} \tag{12}$$

where $k_{P2} = k_{P1} * k_P$, $k_{I2} = k_{I1} * k_I$, $k_{D2} = k_{D1} * k_D k_{P1}$, $k_{I1}$ and $k_{D1}$ are the FL control gains that change in real time with the system's output under control.

The flowchart for implementing the modified INC algorithm with an FL self-tuning-based PID controller is depicted in Fig 6. Utilizing an adaptive controller to track the maximum power and maintain the DC output voltage as environmental circumstances fluctuate is the main objective of this work. This is accomplished by changing the boost PC's D to the voltage level at which the MPP is attained. To determine how well the suggested adaptive controller (INC-FST) keeps up with environmental changes, it is compared to the non-adaptive one (INC-FL). The PVS and two controllers are plotted together in Fig 5.

## 5. Simulation results and discussions

The performance of the PVS to achieve MPPT in terms of PV voltage, current, DC-link voltage, and power under several circumstances of Is and Ts are presented using INC-FST. In addition, a comparison between this proposed MPPT algorithm and P&O, INC, INC-FL is presented. The error and its derivative represented the two inputs of the FL controller. There are 25 rules in the obtained FL inference system's base rules as given in Table 3.

### 5.1 Case 1: Irradiance change at a constant temperature

In this case, the I is increased from 250 to 1000 W/m$^2$ between 1 and 4 s while the T is kept constant at 25°C as seen in Fig 7a. This increase slightly affected the PV power and current as depicted in Fig 7b and 7d, respectively. The best profile with fewer oscillations is obtained when applying the proposed MPPT algorithm, INC-FST. PV voltage is slightly increased with some oscillations as shown in Fig 7c. When utilizing P&O and INC, these oscillations are high, but when using INC-FL and INC-FST, they are low. Due to INC-advantage FSTs over INC-FL, these oscillations are minimal. The DC-link voltage is shown in Fig 7e.

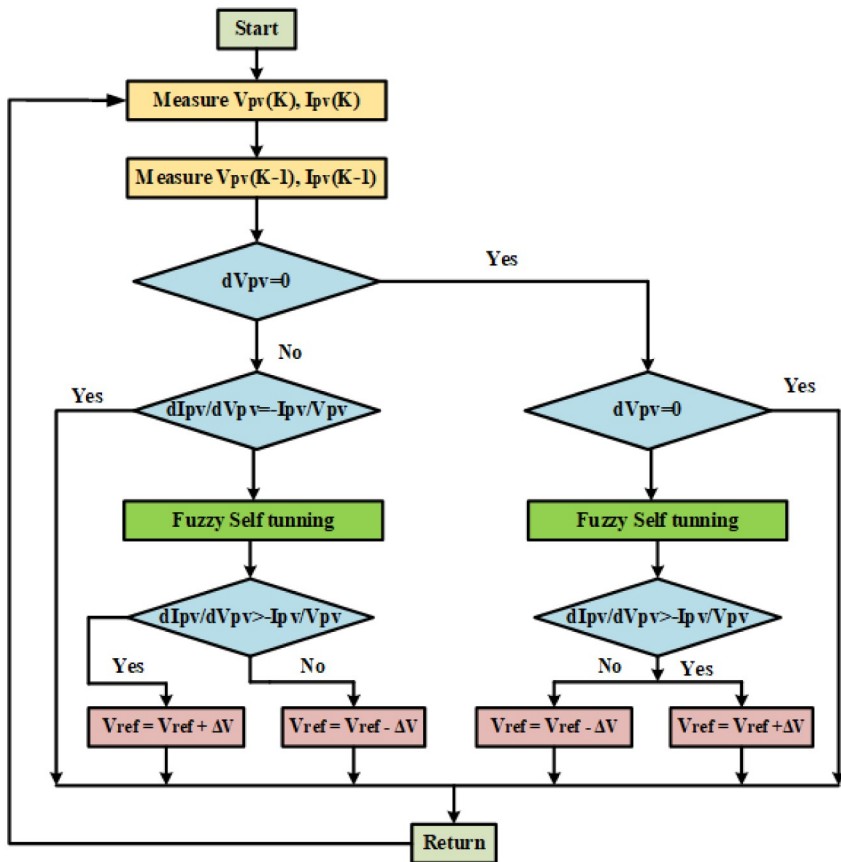

**Fig 6. Flowchart for implementation of the modified INC algorithm with FL self-tuning based PID controller.**

## 5.2 Case 2: Temperature change at a constant irradiance

The I in this instance is maintained at 1000 W/m$^2$ and the T is increased suddenly from 25°C to 50°C between 2 and 3.5s as seen in Fig 8a. This increase results in a reduction in the power extracted from the PV system as in Fig 8b. In comparison to the other three strategies, the proposed INC-FST MPPT technique is able to achieve maximum power (P&O, INC, INC-FL) as shown in Fig 8b. The MPP when using P&O, INC, INC-FL, and INC-FST are 99.85, 99.88, 100, and 100 KW respectively. Moreover, fewer oscillations in the PV power are observed when using INC-FST. The highest value of the PV current was obtained when using the INC-FST algorithm, as shown in Fig 8d. When utilizing INC-FST, the PV voltage profile is

**Table 3. Fuzzy associative matrix.**

| Δ error \ Error | NV | N | Z | P | PV |
|---|---|---|---|---|---|
| **NV** | DZ | DZ | DN | DP | DPV |
| **N** | DZ | DZ | DZ | DP | DPV |
| **Z** | DNV | DN | DZ | DP | DPV |
| **P** | DNV | DN | DZ | DZ | DZ |
| **PV** | DNV | DN | DP | DZ | DZ |

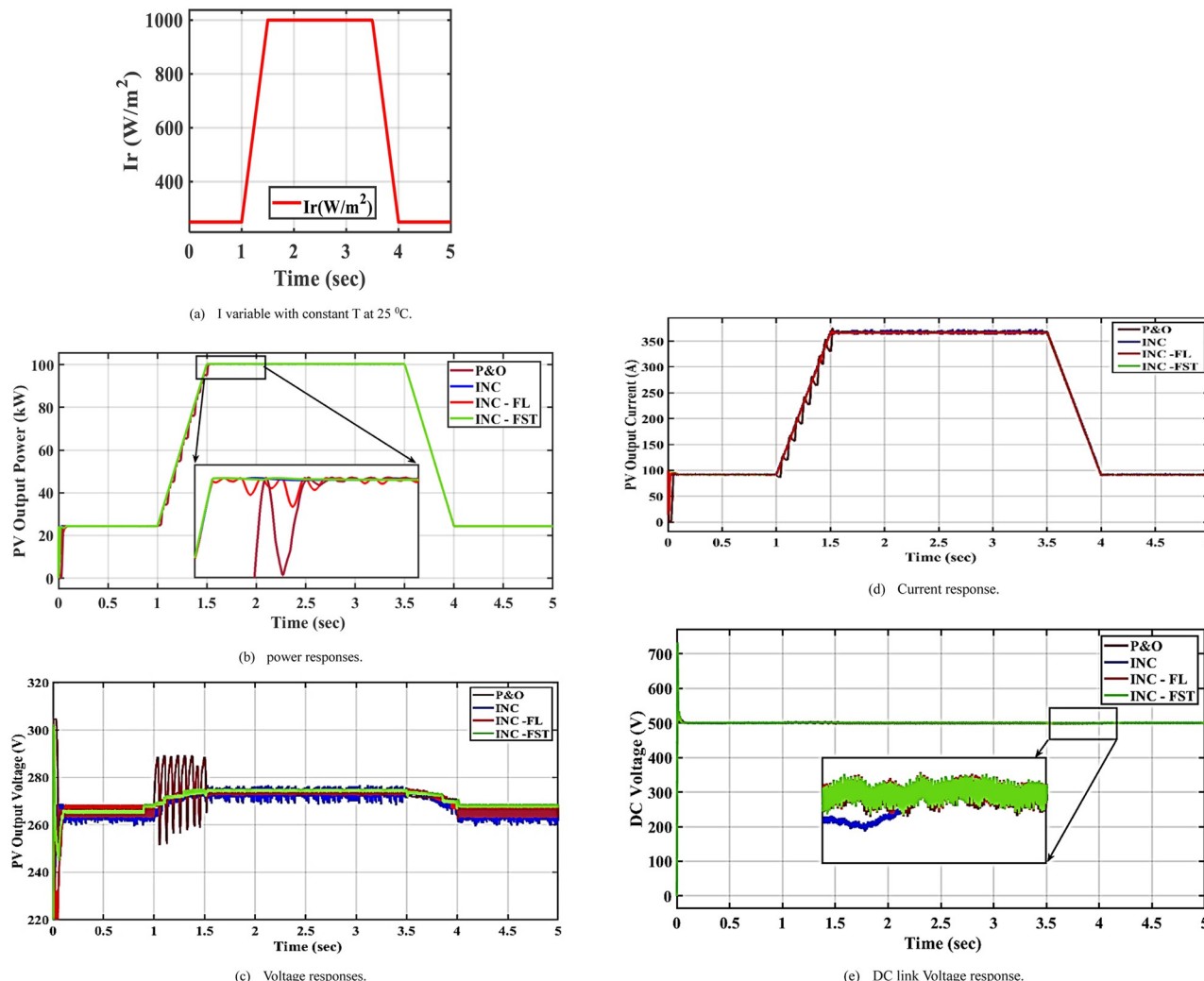

**Fig 7. System performance at constant temperature and changing irradiance.** (a) I variable with constant T at 25 ˚C. (b) power responses. (c) Voltage responses. (d) Current response. (e) DC link Voltage response.

improved with little oscillations, however, there are some oscillations when using the other three MPPT algorithms as depicted in Fig 8c. The DC-link voltage is shown in Fig 8e.

### 5.3 Case 3: Variation in irradiance and temperature

To examine the efficiency of the INC-FST in tracking the MPP with environmental changes, a third test case is introduced. With the changes in T and I values as in test cases 1 and 2, there were abrupt fluctuations in both I and T in this case. At 1s, the PV voltage and power rise as the I does, whereas at 2.5 s, they fall as the T rises as shown in Fig 9b. Fewer oscillations in the PV power are attained when applying INC-FST. Applying the suggested adaptive INC-FST led to the largest PV current with the fewest oscillations among the four MPPT approaches, as demonstrated in Fig 9d. The best PV and DC-link voltage profiles were obtained when using the adaptive MPPT technique, as depicted in Fig 9c and 9e, respectively. In case 3, the effect of the change in radiation with the temperature change is shown, and they are highlighted in Fig 9a. At the time 1 to 1.5 s the change in I and during the time from 2 to 2.5 s the T change.

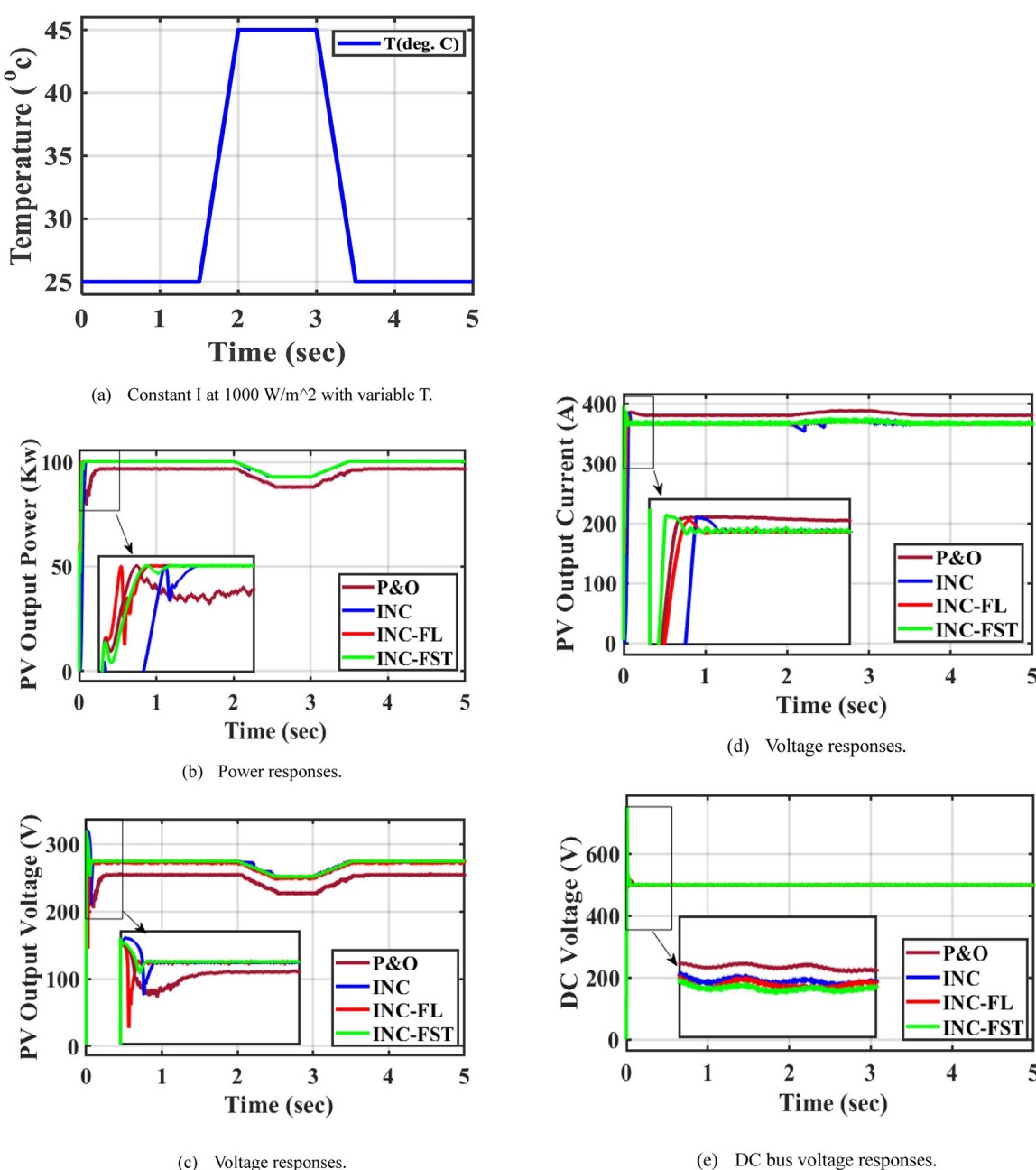

(a) Constant I at 1000 W/m^2 with variable T.

(b) Power responses.

(c) Voltage responses.

(d) Voltage responses.

(e) DC bus voltage responses.

**Fig 8. System performance at constant irradiance and changing temperature.** (a) Constant I at 1000 W/m^2 with variable T. (b) Power responses. (c) Voltage responses. (d) Voltage responses. (e) DC bus voltage responses.

Some control measures for the PV voltage, current, power, and DC-link voltage when using the four MPPT algorithms for all studied cases are summarized in Table 4. According to this comparison, among the four algorithms, the proposed INC-FST achieved the maximum MPP with the best profile for the PV power. The INC-modified algorithm could improve the profile of PV current, voltage, and DC-link voltage. An extensive study of the most recent MPPT methods together with the suggested MPPT strategy is provided in Table 5 to prove the proposed technique's effectiveness. In this table, all high, medium, and low are represented by H,

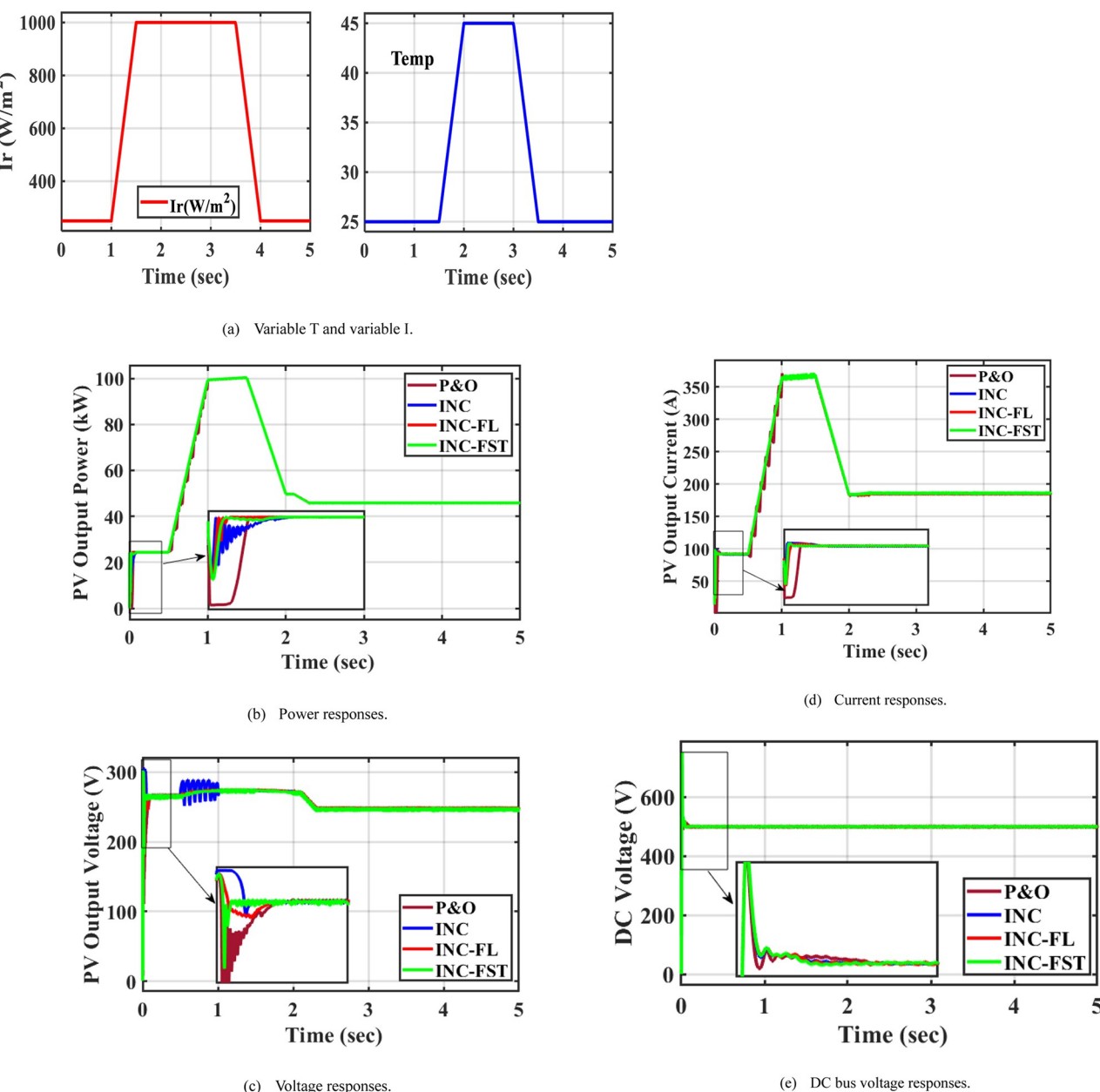

**Fig 9. System performance at irradiance and temperature variations.** (a) Variable T and variable I. (b) Power responses. (c) Voltage responses. (d) Current responses. (e) DC bus voltage responses.

M, and L, respectively. Furthermore, investigated system efficacy under four techniques in all studied cases is presented in Fig 10 to prove the proposed method effectiveness.

## 6. Conclusions

In this study, the FST approach was utilized as an adaptive control technique to modify the gains of the PID controller in the INC. The objective was to regulate the PVS terminal voltage to attain the MPP in grid-connected PVS, particularly in scenarios including rapid and fluctuating changes in the IR and T. The development of the MPPT controller adaption aims to

**Table 4. Control measures for all studied cases.**

| Variable | Control measures | Case. No | P&O | INC | INC-FL | INC-FST |
|---|---|---|---|---|---|---|
| PV output power | MPP (kW) | Case 1 | 99.84 | 99.88 | 100 | 100 |
| | | Case 2 | 99.73 | 99.79 | 99.93 | 99.95 |
| | | Case 3 | 99.69 | 99.74 | 99.86 | 99.92 |
| | Maximum overshoot % | Case 1 | 0.9799 | 0.9623 | 0.7108 | 0.6822 |
| | | Case 2 | 0.9892 | 0.9617 | 0.7114 | 0.6817 |
| | | Case 3 | 0.9888 | 0.9611 | 0.7119 | 0.6811 |
| | Rise time (s) | Case 1 | 0.0481 | 0.0886 | 0.0279 | 0.0236 |
| | | Case 2 | 0.0485 | 0.0889 | 0.0284 | 0.0240 |
| | | Case 3 | 0.0489 | 0.0898 | 0.0292 | 0.0249 |
| | Setting time (s) | Case 1 | 4.9223 | 5.6132 | 5.9121 | 4.0232 |
| | | Case 2 | 4.9227 | 5.6137 | 5.9127 | 4.0237 |
| | | Case 3 | 4.9233 | 5.6221 | 5.9135 | 4.0240 |
| PV output voltage | Maximum Overshoot % | Case 1 | 0.1359 | 0.1251 | 0.1188 | 0.1173 |
| | | Case 2 | 0.1361 | 0.1260 | 0.1193 | 0.1180 |
| | | Case 3 | 0.1344 | 0.1267 | 0.1210 | 0.1177 |
| | Rise time (s) | Case 1 | 0.0483 | 0.0161 | 0.0131 | 0.0049 |
| | | Case 2 | 0.0488 | 0.0167 | 0.0135 | 0.0053 |
| | | Case 3 | 0.0491 | 0.0173 | 0.0139 | 0.0061 |
| | Setting time (s) | Case 1 | 5.7321 | 5.6210 | 4.0124 | 4.0219 |
| | | Case 2 | 5.7329 | 5.6215 | 4.0127 | 4.0222 |
| | | Case 3 | 5.7331 | 5.6220 | 4.0132 | 4.0273 |
| PV output current | Maximum Overshoot % | Case 1 | 0.2215 | 0.0328 | 0.1256 | 0.0180 |
| | | Case 2 | 0.2217 | 0.0330 | 0.1258 | 0.0183 |
| | | Case 3 | 0.2222 | 0.0335 | 0.1260 | 0.0185 |
| | Rise time (s) | Case 1 | 0.0473 | 0.0157 | 0.0227 | 0.0130 |
| | | Case 2 | 0.0475 | 0.0160 | 0.0230 | 0.0131 |
| | | Case 3 | 0.0478 | 0.0163 | 0.0235 | 0.0135 |
| | Setting time (s) | Case 1 | 5.4302 | 5.2132 | 4.0241 | 4.0176 |
| | | Case 2 | 5.4310 | 5.2137 | 4.0244 | 4.0179 |
| | | Case 3 | 5.4315 | 5.2133 | 4.0245 | 4.0180 |
| DC link voltage | Maximum Overshoot % | Case 1 | 0.4506 | 0.4620 | 0.4642 | 0.450 |
| | | Case 2 | 0.4509 | 0.4623 | 0.4647 | 0.456 |
| | | Case 3 | 0.4511 | 0.4630 | 0.4652 | 0.459 |
| | Rise time (s) | Case 1 | 0.0043 | 0.0048 | 0.0045 | 0.0047 |
| | | Case 2 | 0.0046 | 0.0052 | 0.0056 | 0.0049 |
| | | Case 3 | 0.0047 | 0.0056 | 0.0061 | 0.0052 |
| | Setting time (s) | Case 1 | 0.0516 | 0.0475 | 0.0425 | 0.0401 |
| | | Case 2 | 0.0519 | 0.0479 | 0.0427 | 0.0404 |
| | | Case 3 | 0.0521 | 0.0480 | 0.0430 | 0.0406 |
| **Efficiency** | | Case 1 | 98.85% | 98.92% | 99.52% | 99.80% |
| | | Case 2 | 98.82% | 98.91% | 99.51% | 99.76% |
| | | Case 3 | 98.90% | 98.90% | 99.55% | 99.73% |

improve its performance and efficacy in response to environmental changes. The findings demonstrate the effectiveness of the proposed methodology. The effectiveness of the INC-FST variations ranges from 99.73% to 99.80%, whereas the P&O, INC, and INC-FL types exhibit efficacy ranging from 98.82% to 98.90%, 98.90% to 98.92%, and 99.51% to 99.55%,

**Table 5. Comparative performance of the suggested technique with recently published works.**

| Ref. | Year | Publisher | Method | Complexity | Efficacy | Oscillations | Sensed variables |
|---|---|---|---|---|---|---|---|
| [45] | 2021 | Elsevier | Quantized sliding mode | M | 98.90% | Very L | Voltage (V) and current (It) |
| [46] | 2023 | Elsevier | Grey wolf-based PID | M | 99.50% | Very L | V&It |
| [47] | 2020 | MDPI | Genetic-INC | M | 99.09% | Very L | V&It |
| [48] | 2020 | Wiely&Hindawie | A self-constructing Lyapunov NN | M | 98.14% | L | V&It |
| [49] | 2021 | Elsevier | Hybrid whale algorithm-based ANFIS | H | 99.35% | L | I, T, and V |
| [50] | 2022 | Elsevier | Reduced oscillations-based P&O | M | 99.49% | Very L | V&It |
| Current study | | | INC-FST | L | 99.80% | Very L | I, and T |

respectively. When comparing the various alternatives, it can be observed that the INC-FST exhibits the lowest rate of MPP fluctuation, the fastest convergence time, the maximum efficacy, and the least amount of ripple. The INC-FST technique showed enhanced performance in terms of PV power profile, voltage, current, and DC-link voltage when compared to the other three techniques investigated. The main objective of our forthcoming undertaking will be the advancement of an INC-FST-based MPPT algorithm for partial shading conditions. In the future, this research can be extended and refined in the following ways:

1. Real-world implementation: Implement the proposed innovative INC-FST in a physical PV system to validate its performance under practical conditions.

2. Hardware validation: Test the algorithm on actual hardware components and assess its effectiveness in improving the efficiency of MPPT under real-world weather fluctuations.

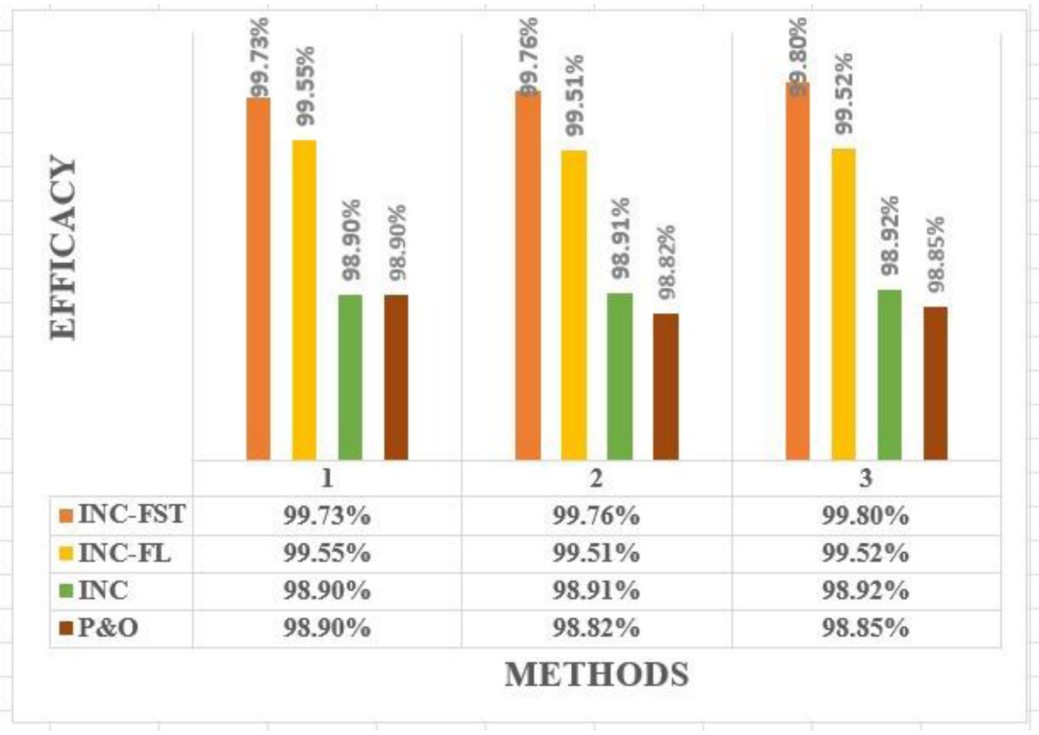

**Fig 10. Investigated system efficacy under four techniques in all studied cases.**

3. Robustness and adaptability: Investigate the robustness of the INC-FST algorithm to various environmental factors beyond current (I) and temperature (T) changes, such as shading, dust, and component degradation.

4. Comparison with emerging techniques: Compare the INC-FST method with emerging MPPT techniques, including artificial intelligence-based algorithms and other state-of-the-art controllers, to assess its competitiveness and advantages.

5. Grid Interaction: Examine the algorithm's behavior when integrated with grid systems, focusing on its impact on power quality, grid stability, and interactions with grid-tied inverters.

6. Cost-Benefit Analysis: Conduct a cost-benefit analysis to evaluate the economic feasibility of implementing the INC-FST algorithm in real-world PV systems, considering factors like initial investment and long-term savings.

## Author Contributions

**Conceptualization:** Daniel Eutyche Mbadjoun Wapet.

**Investigation:** Nagwa F. Ibrahim, Mohamed Metwally Mahmoud, Hashim Alnami, Daniel Eutyche Mbadjoun Wapet.

**Methodology:** Daniel Eutyche Mbadjoun Wapet.

**Resources:** Hashim Alnami, H. Abdelfattah.

**Software:** Nagwa F. Ibrahim, Daniel Eutyche Mbadjoun Wapet.

**Validation:** Mohamed Metwally Mahmoud, Daniel Eutyche Mbadjoun Wapet.

**Writing – original draft:** Nagwa F. Ibrahim, Daniel Eutyche Mbadjoun Wapet.

**Writing – review & editing:** Mohamed Metwally Mahmoud, Hashim Alnami, Daniel Eutyche Mbadjoun Wapet, Sid Ahmed El Mehdi Ardjoun, Mohamed I. Mosaad, Ammar M. Hassan, H. Abdelfattah.

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
