## [Decision Letter · Decision Letter 0]

10 Oct 2023

PONE-D-23-29962A New Adaptive MPPT Technique Using an Improved INC Algorithm Supported by Fuzzy Self-Tuning Controller for a Grid-linked Photovoltaic SystemPLOS ONE

Dear Dr. MBADJOUN WAPET,

Thank you for submitting your manuscript to PLOS ONE. After careful consideration, we feel that it has merit but does not fully meet PLOS ONE’s publication criteria as it currently stands. Therefore, we invite you to submit a revised version of the manuscript that addresses the points raised during the review process.

We look forward to receiving your revised manuscript.

Kind regards,

Praveen Kumar Balachandran

Academic Editor

PLOS ONE

Journal Requirements:

3. Please ensure that you refer to Figure 10 in your text as, if accepted, production will need this reference to link the reader to the figure.

Additional Editor Comments:

Dear Author,

The manuscript needs to be revised as per the reviewers' suggestions.

I suggest adding few more recent references, related to modelling of PV cell, 10.1007/s13198-022-01658-6

Related to comparison of various technoqies, 10.1109/INCET57972.2023.10170183, 10.3390/en15228776

There are lot many recent literature which were not referred, 10.1007/s11356-023-27261-1, 10.3390/en15176172

Reviewers' comments:

Reviewer's Responses to Questions

**Comments to the Author**

1. Is the manuscript technically sound, and do the data support the conclusions?

Reviewer #1: Partly

Reviewer #2: Yes

2. Has the statistical analysis been performed appropriately and rigorously? 

Reviewer #1: No

Reviewer #2: Yes

3. Have the authors made all data underlying the findings in their manuscript fully available?

Reviewer #1: No

Reviewer #2: Yes

4. Is the manuscript presented in an intelligible fashion and written in standard English?

Reviewer #1: Yes

Reviewer #2: Yes

5. Review Comments to the Author

Reviewer #1: 1. The manuscript has some merits, but it lacks a performance comparison with state-of-the-art solutions. Experimental results supporting this comparison are welcome.

2. More experiments should be done to demonstrate the superiority of the proposed method over existing methods.

3. Paper is well written. However, authors should include following papers in literature review section as:

1. DOI:10.1109/JSYST.2018.2817584

2. https://doi.org/10.1049/iet-epa.2017.0804.

3. doi: 10.1109/JSYST.2019.2949083.

4. doi: 10.1109/JSYST.2019.2948899.

5. https://doi.org/10.1049/rpg2.12505

Reviewer #2: 1- Abstract should be improved to reflect novelty and some specific results should be given.

2- Please mention the novelty of this study and suggest further studies.

3- Introduction must be expanded and significantly improved. Novelty is not well reported in the manuscript.

4- Please compare your results and research with the results of other authors and underline what is the scientific novelty in this work.

6. PLOS authors have the option to publish the peer review history of their article (what does this mean?). If published, this will include your full peer review and any attached files.

Reviewer #1: No

Reviewer #2: No

---

## [Author Response · Author response to Decision Letter 0]

13 Oct 2023

***Technical response to the reviewers*** October 14th, 2023

Journal: PLOS ONE

Manuscript No.: PONE-D-23-29962

Title: “A New Adaptive MPPT Technique Using an Improved INC Algorithm Supported by Fuzzy Self-Tuning Controller for a Grid-linked Photovoltaic System” 

Nagwa F. Ibrahim1, Mohamed Metwally Mahmoud2, Hashim Alnami3, Daniel Eutyche Mbadjoun Wapet4, *, Sid Ahmed El Mehdi Ardjoun5, Mohamed I. Mosaad6.7, , Ammar M. Hassan8, and H. Abdelfattah1

1Electrical Department, Faculty of Technology and Education, Suez University, Suez 43533, Egypt

2Department of Electrical Engineering, Faculty of Energy Engineering, Aswan University, Aswan 81528, Egypt

3Electrical Engineering Department, Jazan University, Jazan, 45142, KSA

4*National Advanced School of Engineering, Universit´e de Yaound´e I, Yaound´e, Cameroon

5IRECOM Laboratory, Faculty of Electrical Engineering, Djillali Liabes University, Sidi Bel-Abbes, Algeria

6Electrical & Electronics Engineering Technology Department, Yanbu Industrial College (YIC), Royal Commission Yanbu Colleges & Institutes, Yanbu 46452, Saudi Arabia

7Electrical Engineering Department, Faculty of Engineering, Damietta University, Damietta 34511, Egypt

8Arab Academy for Science, Technology and Maritime Transport, South Valley Branch, Aswan 81516, Egypt.

Nagwa.ibrahim@ind.suezuni.edu.eg, Metwally_M@aswu.edu.eg, halnami@jazanu.edu.sa, eutychedan@gmail.com, m_i_mosaad@hotmail.com, elmehdi.ardjoun@univ-sba.dz, ammar@aast.edu, and hany.abdelfattah@ind.suezuni.edu.eg

*Corresponding author: Daniel Eutyche Mbadjoun Wapet

Dear Editors and Reviewers

The authors are thankful to the learned Editor and Reviewers for their thoughtful and detailed comments to improve the quality of the manuscript. The authors have given reviewer comments a lot of interest in the revision process in an attempt to address all of the reviewers’ concerns and corrections as you will already find them incorporated in the revised manuscript. Moreover, a reply to each of the reviewers’ comments is provided below.

Kindly find the response to the reviewer’s comments in the following paragraphs. We hope this revised version of the manuscript meets the editor and reviewers’ expectations, and the standards of publication in the PLOS ONE Journal. 

The changes carried out by the authors are incorporated in the revised manuscript and highlighted in YELLOW.

Editor's Comments:

Comments to the Authors:

Comment-1: Thank you for submitting your manuscript to PLOS ONE. After careful consideration, we feel that it has merit but does not fully meet PLOS ONE’s publication criteria as it currently stands. Therefore, we invite you to submit a revised version of the manuscript that addresses the points raised during the review process. The paper should be revised thoroughly incorporating the comments offered by reviewers.

Response-1: Our sincere thanks and appreciation to the editor for considering our manuscript for publication in PLOS ONE Journal, and the recommending submission of the revised manuscript with major revisions. To improve the quality of the manuscript, the reviewer's queries are addressed and their suggestions are incorporated into the revised manuscript. The changes carried out by the authors are incorporated in the revised manuscript and highlighted in YELLOW to be easily viewed by the editors and reviewers. A cover letter is provided and prepared to explain, point by point, the details of the revisions to the manuscript. Kindly, check the revised version.

Comment-2: Please include the following items when submitting your revised manuscript: A rebuttal letter that responds to each point raised by the academic editor and reviewer(s). You should upload this letter as a separate file labeled 'Response to Reviewers'. A marked-up copy of your manuscript that highlights changes made to the original version. You should upload this as a separate file labeled 'Revised Manuscript with Track Changes'. An unmarked version of your revised paper without tracked changes. You should upload this as a separate file labeled 'Manuscript'. We look forward to receiving your revised manuscript.

Response-2: Our sincere thanks and appreciation to the editor for his comment. The required items are attached during submission process. A cover letter is provided and prepared to explain, point by point, the details of the revisions to the manuscript. The changes carried out by the authors are incorporated in the revised manuscript and highlighted in YELLOW to be easily viewed by the editors and reviewers. An unmarked version of the revised paper without tracked changes is also provided.

Comment-3: Please ensure that your manuscript meets PLOS ONE's style requirements.

Response-3: The authors are extremely thankful to the editor for this thoughtful point. The revised manuscript meets PLOS ONE's style.

Comment-4: The manuscript needs to be revised as per the reviewers' suggestions.

I suggest adding few more recent references, related to modelling of PV cell, 10.1007/s13198-022-01658-6

Related to comparison of various techniques, 10.1109/INCET57972.2023.10170183, 10.3390/en15228776

There are lot many recent literatures which were not referred, 10.1007/s11356-023-27261-1, 10.3390/en15176172

Response-3: As recommended by the esteemed reviewer, all the suggested recent references are considered. Actually, these references strength the current work. Kindly, check the revised paper. 

Comment-4: Please ensure that you refer to Figure 10 in your text as, if accepted, production will need this reference to link the reader to the figure.

Response-4: The authors are thankful to the esteemed editor upon his valuable comment. All the esteemed editor suggestions are done in the updated paper. 

Reviewers Comments:

Reviewer#1 

Comments to the Authors:

Comment-1: The manuscript has some merits, but it lacks a performance comparison with state-of-the-art solutions. Experimental results supporting this comparison are welcome.

Response-1: First, the authors would like to thank the respected reviewer for his insightful comment and useful feedback that will enhance the presentation and quality of the paper. From our side, we are very keen to answer and take into account the reviewer comment. The proposed technique is investigated and compared with three methods, and a comparison with previously published works is presented in Table 5 in the updated paper. 

The authors would like to emphasize that the application significance of the proposed approach was tested and confirmed using data gathered from a test distribution network. The authors are now working on testing the proposed technique in a Typhoon HIL real-time simulator. The Typhoon HIL real-time simulator has ultra-low-latency, high-high-fidelity, and only single-vendor solutions for real-time application. Typhoon HIL also features very high-resolution (12bit DAC) output pins, allowing the user to obtain highly precise real-time data. Furthermore, the Typhoon HIL operates in a unified environment, ensuring that there are no compatibility concerns at any moment.

In our laboratory, the test bed was in the process of being prepared. Unfortunately, the COVID 19 epidemic and accompanying limitations have prevented this from happening until today. As a result, the authors are constantly working in this direction for the experimental setup, which may be included in future work of the proposed approach. The honorable reviewer reviewer's understanding and cooperation in this regard will be greatly appreciated.

Comment-2: More experiments should be done to demonstrate the superiority of the proposed method over existing methods.

Response-2: The authors are thankful to the honorable reviewer for the words of encouragement and trust in our work. The proposed technique is investigated and compared with three methods, and a comparison with previously published works is presented in Table 5 in the updated paper. Kindly check the revised manuscript.

Table 5. Comparative performance of the suggested technique with recently published works.

Ref. Year Publisher Method Complexity Efficacy Oscillations Sensed variables

[40]

2021 Elsevier Quantized sliding mode M 98.90% Very L Voltage (V) and current (It)

[41]

2023 Elsevier Grey wolf-based PID M 99.50% Very L V&It

[42]

2020 MDPI Genetic-INC M 99.09% Very L V&It

[43]

2020 Wiely&Hindawie A self-constructing Lyapunov NN M 98.14% L V&It

[44]

2021 Elsevier Hybrid whale algorithm-based ANFIS H 99.35% L I, T, and V

[45]

2022 Elsevier Reduced oscillations-based P&O M 99.49% Very L V&It

Current study INC-FST L 99.80% Very L I, and T

Comment-3: Paper is well written. However, authors should include following papers in literature review section as: 1. DOI:10.1109/JSYST.2018.2817584, 2. https://doi.org/10.1049/iet-epa.2017.0804., 3. doi: 10.1109/JSYST.2019.2949083. 4. doi: 10.1109/JSYST.2019.2948899., 5. https://doi.org/10.1049/rpg2.12505

Response-3: The authors thank the reviewer for his recommendation, which would enhance the quality of the paper. The article's references are updated and added the references suggested by reviewer. Kindly refer to the highlighted lines in the reference section.

[16] N. Priyadarshi, S. Padmanaban, P. Kiran Maroti, and A. Sharma, “An Extensive Practical Investigation of FPSO-Based MPPT for Grid Integrated PV System under Variable Operating Conditions with Anti-Islanding Protection,” IEEE Syst. J., vol. 13, no. 2, pp. 1861–1871, 2019, doi: 10.1109/JSYST.2018.2817584.

[17] N. Priyadarshi, S. Padmanaban, M. S. Bhaskar, F. Blaabjerg, and A. Sharma, “Fuzzy SVPWM-based inverter control realisation of grid integrated photovoltaicwind system with fuzzy particle swarm optimisation maximum power point tracking algorithm for a grid-connected PV/wind power generation system: Hardware implementation,” IET Electr. Power Appl., vol. 12, no. 7, pp. 962–971, 2018, doi: 10.1049/iet-epa.2017.0804.

[18] N. Priyadarshi, S. Padmanaban, J. B. Holm-Nielsen, F. Blaabjerg, and M. S. Bhaskar, “An Experimental Estimation of Hybrid ANFIS-PSO-Based MPPT for PV Grid Integration under Fluctuating Sun Irradiance,” IEEE Syst. J., vol. 14, no. 1, pp. 1218–1229, 2020, doi: 10.1109/JSYST.2019.2949083.

[19] N. Priyadarshi et al., “A Hybrid Photovoltaic-Fuel Cell-Based Single-Stage Grid Integration with Lyapunov Control Scheme,” IEEE Syst. J., vol. 14, no. 3, pp. 3334–3342, 2020, doi: 10.1109/JSYST.2019.2948899.

[20] N. Priyadarshi, P. Sanjeevikumar, M. S. Bhaskar, F. Azam, I. B. M. Taha, and M. G. Hussien, “An adaptive TS-fuzzy model based RBF neural network learning for grid integrated photovoltaic applications,” IET Renew. Power Gener., vol. 16, no. 14, pp. 3149–3160, 2022, doi: 10.1049/rpg2.12505.

Reviewer#2 

Comments to the Authors:

Comment-1: Abstract should be improved to reflect novelty and some specific results should be given.

Response-1: At the beginning, the authors are thankful to the honorable reviewer for the words of encouragement and trust in our work. The abstract is improved and have numerical data in the updated version. Kindly check the revised manuscript.

Solar energy, a prominent renewable resource, relies on photovoltaic systems (PVS) to capture energy efficiently. The challenge lies in maximizing power generation, which fluctuates due to changing environmental conditions like irradiance and temperature. Maximum Power Point Tracking (MPPT) techniques have been developed to optimize PVS output. Among these, the incremental conductance (INC) method is widely recognized. However, adapting INC to varying environmental conditions remains a challenge. This study introduces an innovative approach to adaptive MPPT for grid-connected PVS, enhancing classical INC by integrating a PID controller updated through a fuzzy self-tuning controller (INC-FST). INC-FST dynamically regulates the boost converter signal, connecting the PVS's DC output to the grid-connected inverter. A comprehensive evaluation, comparing the proposed adaptive MPPT technique (INC-FST) with conventional MPPT methods such as INC, Perturb & Observe (P&O), and INC Fuzzy Logic (INC-FL), was conducted. Metrics assessed include current, voltage, efficiency, power, and DC bus voltage under different climate scenarios. The proposed MPPT-INC-FST algorithm demonstrated superior efficiency, achieving 99.80%, 99.76%, and 99.73% for three distinct climate scenarios. Furthermore, the comparative analysis highlighted its precision in terms of control indices, minimizing overshoot, reducing rise time, and maximizing PVS power output.

Comment-2: Please mention the novelty of this study and suggest further studies.

Response-2: The authors express their gratitude to the reviewer for providing a valuable comment that will enhance the quality of the paper. In addition, future research direction section is added. Kindly check the revised manuscript.

The paper introduces an innovative INC-FST to improve MPPT efficiency, which will reduce system management complexities and effectively handle instabilities and disturbances in the investigated PVS. This work improves the INC-MPPT algorithm with a fuzzy self-tuning-based PID controller (INC-FST) for regulating the MPPT control of the studied system. This self-tuning algorithm will drive the DC-DC PC D to be updated to track the MPP with changes in I and T. The suggested method is simple, has higher dynamic responsiveness, barely oscillates near MPP, tracks quickly, and performs better in variable weather circumstances. The MATLAB/Simulink is used to investigate the proposed technique. Three tests are applied (variable I with constant T (C1), variable T with constant I (C2), and variable I with variable T (C3)) to assess the proposed effectiveness. The suggested technique efficacy under C1, C2, and C3 are 99.80%, 99.76%, and 99.73%, respectively. The PVS output, including power, voltage, current, and DC bus voltage, shows that the proposed technique performs admirably under a variety of conditions, including I and T fluctuations. The key finding of the suggested work is summarised as follows:

• For PVSs to enable effective MPPT, an innovative INC-FST is suggested.

• The suggested controller has low instabilities around MPP, excellent efficacy, faster rapid responsiveness, and quick converging time.

• The INC-FST is resistant to I and T change since it is adaptable.

• The enhanced algorithm is contrasted with three control approaches (P&O, INC, and INC-FL controller). The improved INC algorithm outperformed previous algorithms in some control indices, such as overshoot, rise time, and settling time, and demonstrated a quicker dynamic reaction for MPPT.

In the future, this research can be extended and refined in the following ways:

1. Real-world implementation: Implement the proposed innovative INC-FST in a physical PV system to validate its performance under practical conditions.

2. Hardware validation: Test the algorithm on actual hardware components and assess its effectiveness in improving the efficiency of MPPT under real-world weather fluctuations.

3. Robustness and adaptability: Investigate the robustness of the INC-FST algorithm to various environmental factors beyond current (I) and temperature (T) changes, such as shading, dust, and component degradation.

4. Comparison with emerging techniques: Compare the INC-FST method with emerging MPPT techniques, including artificial intelligence-based algorithms and other state-of-the-art controllers, to assess its competitiveness and advantages.

5. Grid Interaction: Examine the algorithm's behavior when integrated with grid systems, focusing on its impact on power quality, grid stability, and interactions with grid-tied inverters.

6. Cost-Benefit Analysis: Conduct a cost-benefit analysis to evaluate the economic feasibility of implementing the INC-FST algorithm in real-world PV systems, considering factors like initial investment and long-term savings.

Comment-3: Introduction must be expanded and significantly improved. Novelty is not well reported in the manuscript.

Response-3: The authors thank the reviewer for pointing this point out. kindly check the updated manuscript. The introduction section is expanded and the novelty is presented in the contributions part in the introduction. Kindly check the revised manuscript.

Comment-4: Please compare your results and research with the results of other authors and underline what is the scientific novelty in this work.

Response-4: The authors are thankful to the honorable reviewer for the words of encouragement and trust in our work. The proposed technique is investigated and compared with three methods, and a comparison with previously published works is presented in Table 5 in the updated paper. Kindly check the revised manuscript.

Table 5. Comparative performance of the suggested technique with recently published works.

Ref. Year Publisher Method Complexity Efficacy Oscillations Sensed variables

[40]

2021 Elsevier Quantized sliding mode M 98.90% Very L Voltage (V) and current (It)

[41]

2023 Elsevier Grey wolf-based PID M 99.50% Very L V&It

[42]

2020 MDPI Genetic-INC M 99.09% Very L V&It

[43]

2020 Wiely&Hindawie A self-constructing Lyapunov NN M 98.14% L V&It

[44]

2021 Elsevier Hybrid whale algorithm-based ANFIS H 99.35% L I, T, and V

[45]

2022 Elsevier Reduced oscillations-based P&O M 99.49% Very L V&It

Current study INC-FST L 99.80% Very L I, and T

The authors once again thank the learned Editors and Reviewers for their valuable comments for improving the quality of the manuscript.

---

## [Editor Report · Decision Letter 1]

17 Oct 2023

A New Adaptive MPPT Technique Using an Improved INC Algorithm Supported by Fuzzy Self-Tuning Controller for a Grid-linked Photovoltaic System

PONE-D-23-29962R1

Dear Dr. MBADJOUN WAPET,

We’re pleased to inform you that your manuscript has been judged scientifically suitable for publication and will be formally accepted for publication once it meets all outstanding technical requirements.

Kind regards,

Praveen Kumar Balachandran

Academic Editor

PLOS ONE

Additional Editor Comments (optional):

All the suggested comments were incorporated and addressed properly.

I recommend, the manuscript may be accepted in the present form.
---

## [Editor Report · Acceptance letter]

26 Oct 2023

PONE-D-23-29962R1 

A New Adaptive MPPT Technique Using an Improved INC Algorithm Supported by Fuzzy Self-Tuning Controller for a Grid-linked Photovoltaic System 

Dear Dr. Mbadjoun Wapet:

I'm pleased to inform you that your manuscript has been deemed suitable for publication in PLOS ONE. Congratulations! Your manuscript is now with our production department. 

Kind regards, 

on behalf of

Dr. Praveen Kumar Balachandran 

Academic Editor

PLOS ONE